# Super-radiance reveals infinite-range dipole interactions through a nanofiber

P. Solano[1], P. Barberis-Blostein[1,2], F.K. Fatemi[3], L.A. Orozco[1] & S.L. Rolston[1]

Atoms interact with each other through the electromagnetic field, creating collective states that can radiate faster or slower than a single atom, i.e., super- and sub-radiance. When the field is confined to one dimension it enables infinite-range atom–atom interactions. Here we present the first report of infinite-range interactions between macroscopically separated atomic dipoles mediated by an optical waveguide. We use cold $^{87}$Rb atoms in the vicinity of a single-mode optical nanofiber (ONF) that coherently exchange evanescently coupled photons through the ONF mode. In particular, we observe super-radiance of a few atoms separated by hundreds of resonant wavelengths. The same platform allows us to measure sub-radiance, a rarely observed effect, presenting a unique tool for quantum optics. This result constitutes a proof of principle for collective behavior of macroscopically delocalized atomic states, a crucial element for new proposals in quantum information and many-body physics.

[1] Joint Quantum Institute, Department of Physics and NIST, University of Maryland, College Park, MD 20742, USA. [2] Instituto de Investigaciones en Matemáticas Aplicadas y en Sistemas, Universidad Nacional Autónoma de México, Ciudad Universitaria, México 04510 D.F., Mexico. [3] Army Research Laboratory, Adelphi, MD 20783, USA. Correspondence and requests for materials should be addressed to P.S. (email: solano.pablo.a@gmail.com)

A new class of quantum technologies exploits the interfaces between propagating photons and cold atoms[1–10]. Recent realizations using optical nanofibers (ONFs) platforms include optical isolators, switches, memories, and reflectors[11]. These devices guide the electromagnetic field, a feature that could allow engineering and control a collective time evolution of macroscopically separated subsystems. States that evolve as a whole with dynamics different to that of the independent subsystems are called collective states. These states emerge from atoms interacting via a common mode of the electromagnetic field, and their generation and control can enable aditttional tools for atomic-based technologies[12–18] and the study of many-body physics[19, 20].

For an ensemble of $N$ two-level atoms, in the single excitation limit,

$$|\Psi_\alpha(t)\rangle \propto e^{-\frac{1}{2}(\gamma_\alpha + i\Omega_\alpha)t} \sum_{j=1}^{N} c_{\alpha j}|g_1 g_2 \cdots e_j \cdots g_N\rangle \tag{1}$$

represents the $\alpha$-th collective state of the system, where $\gamma_\alpha$ and $\Omega_\alpha$ are its collective decay and frequency shift, respectively, and $\sum_{j=1}^{N}|c_{\alpha j}|^2 e^{-\gamma_\alpha t}$ is the probability of having an excitation in the atoms. When $\gamma_\alpha$ is larger (shorter) than the natural radiative decay time $\gamma_0$, the system is super- (sub-)radiant[21, 22]. For free space coupling, collective states emerge for atom–atom separations smaller than a few wavelengths[23]. By externally exciting the atoms, super-radiant states are readily observed, but because sub-radiant states are decoupled from the electromagnetic vacuum field, they are challenging to produce[24].

The master equation that describes the dynamics of an ensemble of atomic dipoles, of density matrix $\rho$, coupled through the electromagnetic field is given by ref. [25]

$$\dot{\rho} = -i[H_{\text{eff}}, \rho] + \mathcal{L}[\rho]. \tag{2}$$

The effective Hamiltonian $H_{\text{eff}}$ of the dipolar interaction between atoms and the Lindblad super operator $\mathcal{L}$ in Eq. (2) modify two atomic properties: the resonance frequency and the spontaneous decay rate, respectively. They are given by

$$H_{\text{eff}} = \frac{1}{2}\sum_{i,j}\hbar\Omega_{ij}\sigma_i^\dagger\sigma_j, \tag{3}$$

$$\mathcal{L}[\rho] = \frac{1}{2}\sum_{i,j}\hbar\gamma_{ij}\left(2\sigma_j\rho\sigma_i^\dagger - \sigma_i^\dagger\sigma_j\rho - \rho\sigma_i^\dagger\sigma_j\right), \tag{4}$$

with $\sigma_i$ $\left(\sigma_i^\dagger\right)$ being the atomic lowering (raising) operator for an excitation of the $i$-th atom. $\Omega_{ij}$ is the rate of photons exchanged between atoms and $\gamma_{ij}$ is the term responsible for collective radiative decays, where $\gamma_{ii}$ is the single atom decay rate. The decay of an excitation in such a system, that leads to a collective state as in Eq. (1), depends on the coupling amplitudes and relative phase between the atoms given by $\gamma_{ij}$.

When atoms are far apart in free space, their interaction is mediated by a propagating field with an expanding wavefront, and a separation of few wavelengths is enough to make the interaction negligible. As atoms get closer together, $\Omega_{ij}$ in Eq. (3) diverges, reducing the coherence of a system with more than two atoms. These constraints can be circumvented by using longer wavelengths with larger atomic dipole moments, such as Rydberg atoms[26], or long-range phonon modes, implemented with trapped ions[27, 28]. However, these techniques are limited to sub-wavelength distances. When the field is confined to one

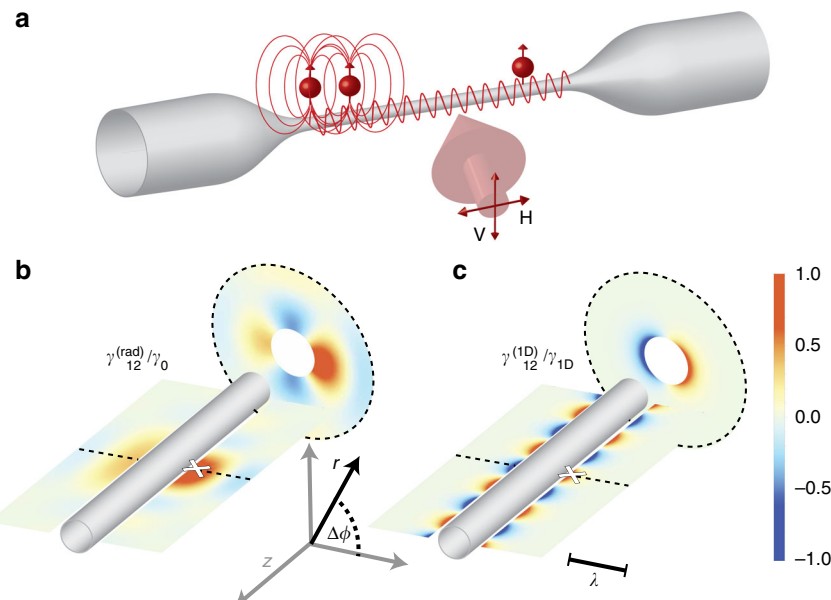

**Fig. 1** Position-dependent atom–atom coupling along the optical nanofiber. **a** Schematic of an ONF as a platform for generating single photon collective atomic states, excited from the side by a weak probe of polarization V or H. When two atoms are close together, the dipolar interaction is mostly mediated by the modes of the electromagnetic field radiating outside the nanofiber. This is a limited-range interaction that decays inversely with distance. When the atoms are widely separated, the guided mode of an ideal ONF mediates the interaction for arbitrary distances. **b, c** Show the atom–atom interaction rate $\gamma_{12}$ (see Eq. (4)) experienced by an atom around the fiber given another atom at the position denoted by the white cross (see "Methods" section for the details of the calculation). Its amplitude is shown for a longitudinal and a transversal cut (specified by dashed black lines). Both plots share the color scale, but in **b** the interaction rate is normalized by the single atom total decay rate $\gamma_0$ and in **c** by the decay rate into the guided mode $\gamma_{1D}$. Along the $z$-axis, the interaction among atoms through free space radiation modes decreases as $\gamma_{12}^{(\text{rad})} \propto \sin(k|\Delta z|)/k\Delta z$ (with $k$ being the wavenumber and $\Delta z$ the separation between two atoms). The infinite interaction through the ONF-guided mode changes as $\gamma_{12}^{(1D)} \propto \cos(\beta_0\Delta z)\cos(\Delta\phi)$ (with $\beta_0$ being the propagation constant of the resonant-guided mode and $\Delta\phi$ the angle difference in cylindrical coordinates). The wavelength $\lambda$ sets the scale in **b, c**

dimension, it enables infinite-range interactions. This has been observed for atoms in an optical cavity[29, 30].

Waveguides offer an alternative by confining the mediating field, where the extent of the interactions is not limited by the cavity size and the field can propagate unaltered for a broad range of frequencies[31, 32], facilitating the coupling of atoms separated by many wavelengths (see Fig. 1). Dipole–dipole interactions, given by $\Omega_{ij}$, are finite for atoms along the waveguide, removing a practical limit for creating super-radiant states of a large number of atoms. Super-radiance of atoms around a waveguide has been observed[7], but its long-range interaction feature has not been proven or explored. Such effect has been implemented with superconducting waveguides and two artificial atoms one wavelength apart[33], but has not been realized for many atoms at multi-wavelength distances in the optical regime.

We present the implementation of collective atomic states through infinite-range interactions via a one-dimensional nanophotonic waveguide. We use a few atoms evanescently coupled to a single-mode ONF, observing super- and sub-radiant radiative decays of a single excitation in the system, evidence of collective behavior. Atoms around the ONF interact at short and long distances (see Fig. 1a), the latter mediated by the ONF-guided mode. The dipolar interaction that leads to a collective decay is separated into two contributions of the electromagnetic field:

from modes radiating outside the ONF, $\gamma_{12}^{(\text{rad})}$, and from the guided mode, $\gamma_{12}^{(1D)}$[25] (see Fig. 1b, c). In particular, we observe sub-radiant decay rates of proximal atoms interacting through the radiated modes and super-radiant decay rates of atoms interacting through the guided mode over distances of hundreds of resonant wavelength.

## Results

**Experimental setup**. We overlap a cold atomic cloud of $^{87}$Rb atoms from a magneto-optical trap (MOT) with a 240 nm radius ONF. This ONF is single mode at the D2 resonant wavelength of 780 nm. After the MOT is turned off, the atoms form a cold thermal gas around the ONF. They are prepared in the $F = 1$ ground level by an external free propagating beam. A repumper beam driving the $F = 1 \rightarrow F = 2$ transition propagates through the nanofiber, leaving in the $F = 2$ ground state-only atoms that interact with the ONF-guided mode. By detuning the repumper below resonance, we address atoms near the nanofiber (whose levels have been shifted by van der Waals interactions) such that the atomic density distribution peaks at ~30 nm away from the surface. A weak free space probe pulse, propagating perpendicular to the fiber, excites atoms for 50 ns using the $F = 2 \rightarrow F' = 3$ transition. After the probe turns off (extinction ratio better than

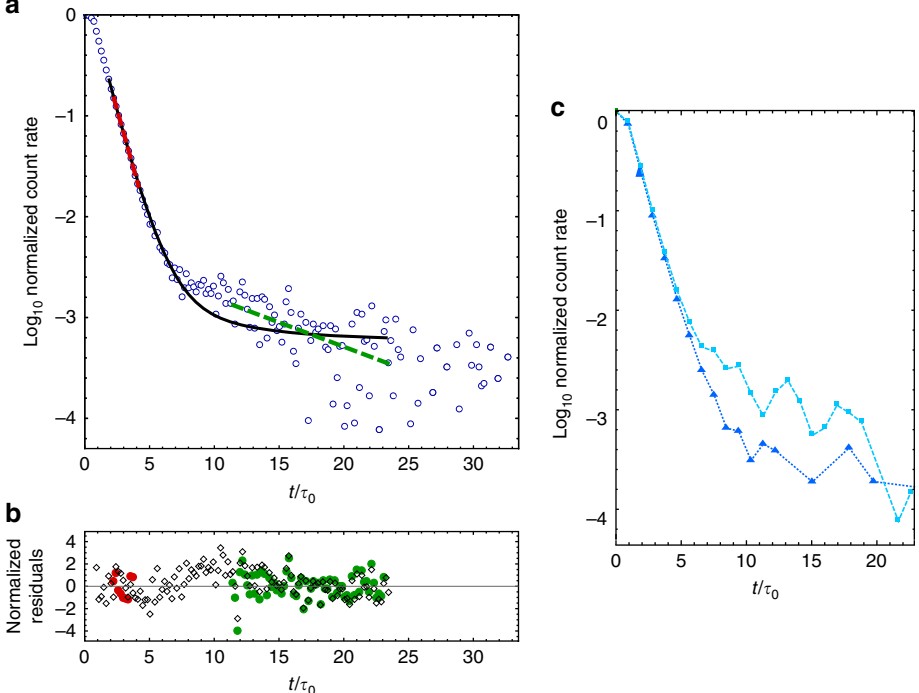

**Fig. 2** Measured super- and sub-radiant decay of excited atoms near the optical nanofiber. **a** Normalized rate of photons detected through the ONF mode (blue circles in a logarithmic scale) as a function of time in units of natural lifetime ($\tau_0 = 1/\gamma_0 = 26.24$ ns) with 5 ns bins. The signal is taken after a probe beam polarized along the nanofiber turns off. In this realization OD = 0.66 ± 0.05. The individual statistical error bars are not plotted but they are taken into account for the normalized residuals in **b**. The number of counts at $t = 0$ exceeds $10^6$. We see two distinct slopes (red and green), at short and long times. The initial slope (red) deviates toward decay rates faster than $\gamma_0$, a signature of super-radiance. The second slope (green) comes from the natural post-selection of purely sub-radiant states. The red dashed (green dashed) line is the best fit to a pure exponential decay of the initial (final) decay. The decay rate of the fit at short times is 1.10 ± 0.02 $\gamma_0$, and 0.13 ± 0.01 $\gamma_0$ for the fit at longer times, with one-sigma error. The one-sigma fractional systematic errors are ±0.01. The full description of the measured temporal evolution of the system involves averaging over many different decay rates through Monte Carlo methods (explained in "Methods" section). The solid black line is a simulation of 7 atoms along the ONF, with reduced $\chi^2$ of 1.60. **b** The red circles, green circles, and black diamonds are the normalized residuals of the exponential fits to the initial decay, final decay, and the theoretical model. **c** Shows two different decay signals from an excitation driving the atoms with light polarized along (cyan rectangles) and perpendicular (blue triangles) to the ONF for 25 ns bins. When the driving field is polarized along the ONF, we observe super- and sub-radiance, and when it is polarized perpendicular to the ONF the super-radiance increases and the sub-radiance decreases. This feature is qualitatively captured by the theoretical model

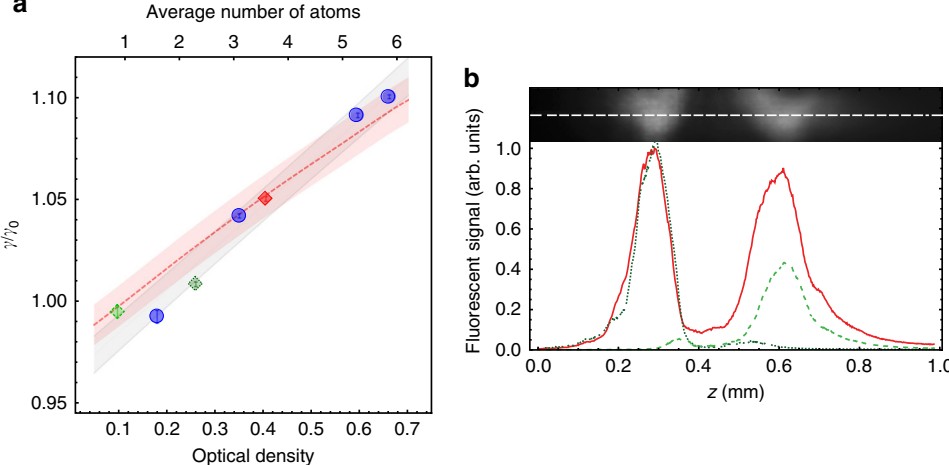

**Fig. 3** Super-radiant decay as a function of atom number including separated clouds. **a** Relationship of the decays as a function of average number of atoms (OD) along the optical nanofiber. The normalized fast decay rates are plotted as a function of the OD (lower abscissa) and $N$ (upper abscissa) measured through the ONF-guided mode. The blue circles correspond to the signals from a single cloud of atoms. We split the atomic cloud in two (as shown in **b**). The dashed light and dotted dark green diamonds, and the solid red square correspond to the right, left, and the combination of both atomic clouds, respectively. The systematic errors (not shown) are estimated to be 1% for the decay rates and smaller than 20% for the atom number. The plotted error bars represent the statistical uncertainty of the fitting to an exponential decay. The gray region represents the one-sigma confidence band of a linear fit to the data. The red dashed line is the theoretical prediction, and the red shaded region represents a confidence interval set by a fractional error of 1%. The curve goes below $\gamma/\gamma_0 = 1$ because the natural decay rate is modified given the geometry of the ONF and the alignment of the atomic dipoles (Purcell effect)[36]. **b** Separated atom clouds show long-range interactions. The top of the figure shows in black and white a fluorescence image of a split MOT. The white dotted line represents the ONF location. The fluorescence signal of the split MOT along the nanofiber is plotted as a function of position. The dashed light (dotted dark) green dashed lines is the intensity distribution of the right (left) atomic cloud when the other one is blocked. The solid red line is the intensity distribution when both clouds are present. The separation between the center of both clouds is $318 \pm 1\,\mu\text{m}$, given by standard error of the mean of a Gaussian fit. This distance is equivalent to 408 wavelengths

$1{:}2 \times 10^3$ in one atomic natural lifetime), we collect photons spontaneously emitted into the ONF mode to measure the decay time using time-correlated single photon counting.

Collective states can be tailored by positioning the atoms in a particular arrangement. This kind of control has been challenging to implement for atoms trapped close enough to the ONF (tens of nanometers) to ensure significant mode coupling. However, collective states are still observed when atoms from a MOT are free to go near the ONF. Their random positioning leads to probabilistic super- or sub-radiant states on each experimental realization. Sub-radiant states have lifetimes much longer than most other processes, favoring their observation. Super-radiance can be measured as an enhanced decay rate at short times. Both effects can provide quantitative experimental evidence of collective states.

**Observation of super- and sub-radiance.** Figure 2 shows a typical signal of the atomic decay as measured through the ONF. Its time dependence can be described by two distinct exponential decays. The slow decay (green dashed line in Fig. 2a) corresponds to an average of sub-radiant decays due to pairs of atoms located within a wavelength, i.e., free space interaction (Fig. 1b). Infinite-range interactions also produce sub-radiant decay rates. However, these events are obscured by the dominant signal of slower decays produced from free space interactions. In our case $\gamma_{1D} \approx 0.13\gamma_0$, so sub-radiance from infinite-range interactions is limited to $\gamma_0 - \gamma_{1D} \approx 0.87\gamma_0$. This is a factor of six faster than the observed sub-radiant rates (green dashed line in Fig. 2a). Sub-radiance of atoms interacting in free space has been observed in a very optically dense cloud of atoms[24], but we can observe it even for optical densities (OD) as small as 0.3. The fast decay rate (red dashed line in Fig. 2a) is larger than the natural decay rate, showing the presence of super-radiant initial states.

A full description of the temporal evolution of the entire data sample requires numerical (Monte Carlo) methods, as the solid black line in Fig. 2 shows. We use the average number of atoms ($N$) as the only free parameter for this simulation, allowing for variations of the background up to one sigma. The two-sigma deviation between simulation and data (see Fig. 2b from 7 to 15 $\tau_0$) could come from otherwise a longer living sub-radiant state that gets prematurely destroyed because atoms fall onto the ONF, emitting the excitation into the guided mode. The initial state preparation—the polarization of the incoming pulse that produces the collective one-photon state—can favor super- or sub-radiant states, as Fig. 2c shows. In general, the free space atom–atom coupling is larger for dipoles driven along the ONF ($z$ in the direction set in Fig. 1b), favoring sub-radiance, and the ONF-mediated coupling is larger for dipoles driven perpendicular to the ONF, favoring super-radiance.

An important difference between sub- and super-radiant decay rates in ONF is that the latter increases as a function of $N$. We can vary $N$ from one to six by changing the MOT density, and quantify it through the OD of the ONF mode. $n_{\text{eff}}\text{OD} = N\gamma_{1D}/\gamma_0$, where $n_{\text{eff}}$ is the mode effective refractive index, and in our case $n_{\text{eff}} \approx 1.15$. We measure the transmission spectrum through the ONF to extract the OD. The decay rate increases with $N$, as shown by the blue circles in Fig. 3, indicating super-radiance. The gray region represents the one-sigma confidence bands of a linear fit to the data showing a linear dependence of the super-radiant decay rate for increasing $N$. The theoretical model implemented for the fit shown in Fig. 2 (solid black line) also predicts a linear dependence on $N$ of the decay rate $\gamma$ at short times. The red dashed line in Fig. 3a shows this prediction, corroborating the theory with the experiment.

**Evidence of infinite-range interactions.** The average spacing between atoms is larger than a wavelength for most of the

realizations, meaning that infinite-range interactions are always present. However, to provide an unambiguous proof of infinite-range interactions, we split the atomic cloud in two (see Fig. 3b). We see that two atomic clouds separated by more than 400 wavelengths present the same super-radiant collective behavior as a function of the OD as a single atomic cloud. This shows that the relevant parameter is the total OD (or $N$) along the ONF mode, regardless the separation between atoms.

## Discussion

Optically guided modes can be used to mediate atom–atom interactions, creating macroscopically delocalized collective atomic states. We use the super-radiant behavior of distant atoms as evidence of infinite-range interaction, but other interesting collective quantum properties remain to be tested. The practical limits of infinite-range interactions are an open question, since in principle optical fibers can be easily connected and rerouted along several meters. An intriguing next step is the study of quantum systems beyond the Markov approximation, coupling atoms at distance greater than what light travels in an atomic lifetime. Moreover, by achieving fine control on the positioning of the interacting particles, and/or using the directional coupling produced by chiral atom–light interaction[10], one can engineer desired states tailored to address specific applications. The implementation of infinite-range interactions opens new possibilities for quantum technologies and many-body physics. Given the application of one-dimensional waveguides in photonic-based quantum technologies, we envision infinite-range interactions as the natural next step toward interconnecting quantum systems on scales suitable for practical applications.

## Methods

**Experimental methods**. A tapered single mode ONF, with waist of $240 \pm 20$ nm radius and 7 mm length, is inside an ultrahigh vacuum (UHV) chamber, where it overlaps with a cloud of cold $^{87}$Rb atoms (less than half a millimeter width) created from a MOT. The MOT is loaded from a background gas produced by a $^{87}$Rb dispenser. Acousto optic modulators (AOMs) control the amplitude and frequencies of the MOT beams. After the atomic cloud loading reaches steady state, the MOT beams are extinguished. A free space propagating depump beam, resonant with the $F = 2 \rightarrow F' = 2$ transition (150 μs duration) prepares all atoms in the cloud in the $F = 1$ ground state. A 0.4 nW fiber-repump beam, detuned below resonance by 15 MHz to the $F = 1 \rightarrow F' = 2$ transition, propagates through the ONF during the entire cycle. It pumps back to the $F = 2$ ground state only those atoms close enough to the ONF to interact with the guided mode. This detuning repumps only those atoms close enough to the ONF surface to experience an energy shift due to the van der Waals interaction with the dielectric body. This produces a narrow density distribution of atoms of 5 nm width centered around 30 nm away from the surface. We wait 300 μs until the AOMs reach maximum extinction. The atomic cloud free falls and expands around the ONF for 2.5 ms creating a cold thermal gas (~150 μK), where each atom interacts with the nanofiber mode for ~1.5 μs[34]. The atomic density reduction due to the cloud expansion limits the probing time of the cycle. The atoms are excited by pulses of a weak probe beam incident perpendicularly to the nanofiber (see Fig. 1a) and linearly polarized along the ONF for the data set shown in Fig. 3. The pulses are resonant with the $F = 2 \rightarrow F' = 3$ transition of the D2 line and created with a double-passed Pockels cell (Conoptics 350–160), with a pulse extinction ratio better than 1:2000 in one atomic natural lifetime that remains at least an order of magnitude below the atomic decay signal for more than 20 lifetimes. The on–off stage of the light pulses is controlled with an electronic pulse generator (Stanford Research Systems DG645). The probe power is kept low, i.e., saturation parameter $s < 0.1$, to ensure a single photon excitation while staying in the limit of low excitation and avoiding photon pileup effects. Only those atoms that interact with the ONF-guided mode are in the $F = 2$ ground state and will be excited by the probe beam. During the probing time, we send a train of 50 ns probe pulses every 1 μs. The probe is a 7 mm $1/e^2$ diameter collimated beam. After 2 ms of probing (~2000 pulses), the probe beam is turned off and the MOT beams are turned back on. During the probing time, the atomic density remains constant. We wait 20 ms after the MOT reloads and repeat the cycle. The average acquisition time for an experimental realization is around 5 h, giving a total of about $1 \times 10^9$ probe pulses. The photons emitted into the nanofiber and those emitted into free space are independently collected with avalanche photodiodes (APDs, laser components COUNT-250C-FC, with less than 250 dark counts per second). The TTL pulses created from photons detected by APD are processed with a PC time-stamp card (Becker and Hickl DPC-230) and time

stamped relative to a trigger signal coming from the pulse generator. We use time-correlated single photon counting[35] to extract the decay rate of a single excitation in the system, eliminating after-pulsing events from the record.

When atoms are around the nanofiber, they tend to adhere due to van der Waals forces. After a few seconds of having the ONF exposed to rubidium atoms it gets coated, suppressing light propagation. To prevent this, we use 500 μW of 750 nm blue-detuned light (Coherent Ti:Sapph 899) during the MOT-on stage to create a repulsive potential that keeps the atoms away from the ONF surface. This is intense enough to heat the ONF and accelerate the atomic desorption from the surface. The blue-detuned beam is turned off at the same time as the MOT beams, so the probed atoms are free to get close to the nanofiber.

Photons from the probe beam can be scattered multiple times by the atoms producing a signal that looks like a long decay, an effect known as radiation trapping. This effect can obscure sub-radiant signals. However, the small ODs involved in the experiment allow us to neglect contributions from radiation trapping. We confirm this assumption by observing the same temporal evolution of the signal at constant OD for several detunings of the probe beam in a range of ±3 linewidths[24].

The atomic lifetime can also be altered by modification of the electromagnetic environment of the atoms in the presence of an ONF, i.e., the Purcell effect. However, this effect is characterized separately[36] and well understood. More importantly, it does not depend on the number of atoms, in contrast with the super-radiant behavior.

Further evidence of collective states can be found in the resonance spectrum of the system (see Eqs. (2) and (3)). The dispersive part of the interaction modifies the resonance frequencies of the system, due to avoiding crossing of otherwise degenerate levels. This effect is in principle visible in the transmission spectrum. In our particular case, the frequency splitting is a small percentage of the linewidth. Broadening mechanisms and other systematic errors prevent us from clearly observing such signal. However, a line-shape dependence on $N$ can be inferred from the statistical analysis of the fit of the spectrum to a Lorentzian. This effect might enable the exploration of features of collective states in the spectral domain.

ONFs can provide chiral atom–light coupling[10]. Even though this is a promising feature of the platform, it requires a particular positioning of the atoms and a preparation of their internal state. This first exploration of infinite-range interactions involves detecting only on one end of the ONF and azimuthally averaging the atomic position, preventing studies of chiral effects that we do not consider crucial to our measurements.

**Theoretical model**. We follow the work of Svidzinsky and Chang[37] to implement the theoretical simulations of the experiment. Consider the Hamiltonian of $N$ atoms interacting with an electromagnetic field in the rotating-wave approximation

$$\hat{H}_{\text{int}} = -\sum_k \sum_{j=1}^N \hbar G_{kj} \left[ \hat{\sigma}_j \hat{a}_{\mathbf{k}}^\dagger e^{i(\omega - \omega_0)t} + h.c. \right] \tag{5}$$

where $\hat{\sigma}_j$ is the lowering operator for atom $j$; $\hat{a}_{\mathbf{k}}^\dagger$ is the photon creation operator in the mode $k$-th; $\omega_0$ and $\omega$ are the frequencies of atomic resonance and $k$-th mode of the field, respectively. This is a general expression for the Hamiltonian, which leads to the master equation in Eq. (2) after some approximations. The sum on $j$ is done over the atoms and the sum on $k$ goes over the electromagnetic field modes, guided into the nanofiber and radiated outside. These modes can be found in the work of Le Kien et al.[25]. The sum over the guided modes is $\sum_\mu = \sum_{f,p} \int_0^\infty d\omega$, where $f$ and $p$ are the propagation direction and polarization (in the circular basis (plus or minus)) of the guided mode, respectively, and $\mu$ stands for modes with different parameters $(\omega, f, p)$. The sum over the radiated modes is $\sum_\nu = \sum_{m,p} \int_0^\infty d\omega \int_{-k}^k d\beta$; where $m$ is the mode order, $k$ is the wavenumber, $\beta$ is the projection of the wave vector along the fiber or propagation constant, and $\nu$ stands for modes with different parameters $(\omega, \beta, m, p)$. Then the total sum is $\sum_k = \sum_\mu + \sum_\nu$. The electromagnetic field modes and their relative coupling strength have been previously studied[25]. The coupling frequencies $G_{kj}$ for the guided and radiated modes can be written as:

$$G_{\mu j} = \sqrt{\frac{\omega \beta'}{4 \pi \epsilon_0 \hbar}} \left[ \mathbf{d}_j \cdot \mathbf{e}^{(\mu)} \left( r_j, \phi_j \right) \right] e^{i(f \beta z_j + p \phi_j)} \tag{6}$$

$$G_{\nu j} = \sqrt{\frac{\omega}{4 \pi \epsilon_0 \hbar}} \left[ \mathbf{d}_j \cdot \mathbf{e}^{(\nu)} \left( r_j, \phi_j \right) \right] e^{i(\beta z_j + m \phi_i)} \tag{7}$$

where $\beta' = d\beta/d\omega$, $\mathbf{d}_j$ is the dipole moment of the $j$-th atom, and $\mathbf{e}^{(\mu,\nu)}$ are the electric field profile function (or spatial dependence of the amplitude) of the guided and radiated modes ($\mu$ and $\nu$).

Atoms interact with each other mediated by the electromagnetic field. The interaction between the atomic dipoles is proportional to the product of the atom–light coupling frequencies of the form $G_{ki}G_{kj}$, where $k$ labels the mediating field mode (the repetition of the letter implies summation if there is more than one mode) and $i$ and $j$ label the $i$-th and $j$-th atom. It is possible to identify two contributions from the coupling of atoms to the dynamics of the system, a

dispersive and a dissipative one, as shown in Eq. (2). The dispersive part contributes to the unitary evolution of the system (see Eq. (3)), and it can be decomposed as $\Omega_{ij} = \Omega_{ij}^{(\mathrm{rad})} + \Omega_{ij}^{(1\mathrm{D})}$, where $\Omega_{ij}^{(\mathrm{rad})}$ and $\Omega_{ij}^{(1\mathrm{D})}$ come from the interaction of the $i$-th and $j$-th atoms mediated by the radiated and guided modes, respectively. $\Omega_{ij}$ is usually called the dipole–dipole coupling frequency. The dissipative part contributes to the decay of the system (see Eq. (4)), and it can be decomposed as $\gamma_{ij} = \gamma_{ij}^{(\mathrm{rad})} + \gamma_{ij}^{(1\mathrm{D})}$, where $\gamma_{ij}^{(\mathrm{rad})}$ and $\gamma_{ij}^{(1\mathrm{D})}$ come from the interaction of the $i$-th and $j$-th atoms mediated by the radiated and guided modes, respectively. For simplicity, here we focus only on the case where atoms are regarded as two-level systems prepared in an initial state with induced atomic dipoles aligned along the ONF ($z$-axis). This is a reasonable approximation for atoms weakly driven by an external probe polarized along $z$. In a realistic scenario, the light scattered by the fiber and by the multi-level internal structure of the atoms can mix the light polarization. The computation of such a system becomes cumbersome and only contributes to correction to the dominant effect. A description given by two-level atoms aligned along the $z$-axis allows us to quantitatively capture the physical phenomena while keeping the mathematical description simple. For atoms placed in the position $\mathbf{r}_i = (r_i, \phi_i, z_i)$ with reduced dipole moment $d_i$, we obtain

$$\gamma_{ij}^{(1\mathrm{D})} = \frac{2\omega_0 \beta_0'}{\epsilon_0 \hbar} d_i d_j e_z^{(\mu_0)}(r_i) e_z^{*(\mu_0)}(r_j) \cos\left(\phi_i - \phi_j\right) \cos\beta_0(z_i - z_j) \tag{8}$$

$$\gamma_{ij}^{(\mathrm{rad})} = \frac{2\omega_0}{\epsilon_0 \hbar} d_i d_j \sum_m \int_0^{k_0} \mathrm{d}\beta\, e_z^{(\nu)}(\mathbf{r}_i) e_z^{*(\nu)}(\mathbf{r}_j) \times \cos\mathrm{m}\left(\phi_i - \phi_j\right) \cos\beta(z_i - z_j) \tag{9}$$

$$\Omega_{ij}^{(1\mathrm{D})} \approx \frac{\omega_0 \beta_0'}{\epsilon_0 \hbar} d_i d_j e_z^{(\mu_0)}(r_i) e_z^{*(\mu_0)}(r_j) \cos\left(\phi_i - \phi_j\right) \sin\beta_0(z_i - z_j) \tag{10}$$

where $\mu_0$ parametrizes the guided modes on resonance. The dispersive component of the interaction given by the radiated modes as $\Omega_{ij}^{(\mathrm{rad})}$ is a complicated expression and hard to solve even numerically. We follow the work of Le Kien et al.[38] and use the free space value of $\Omega_{ij}^{(\mathrm{rad})}$ throughout the calculation as a reasonable approximation. $\gamma_{ii} = \gamma_0$ with $\gamma_0$ the single atom natural decay rate. $\gamma_{12}^{(\mathrm{rad})}$ and $\gamma_{12}^{(1\mathrm{D})}$ are plotted in Fig. 1b, c, respectively, for an atom fixed at $\mathbf{r}_1 = (240 + 30)$ nm, $0$, $0$) (240 nm being the ONF radius and 30 nm the distance of the atom to the surface). When atoms are too close to each other, the radiated terms $\Omega_{ij}^{(\mathrm{rad})}$ and $\gamma_{ij}^{(\mathrm{rad})}$ dominate over the guided ones ($\Omega_{ij}^{(1\mathrm{D})}$ and $\gamma_{ij}^{(1\mathrm{D})}$), with $\Omega_{ij}^{(\mathrm{rad})}$ diverging and $\gamma_{ij}^{(\mathrm{rad})}$ approaching the total decay rate. With a low number of atoms randomly distributed along the ONF, the effects of short-range interaction are small but still observable.

For simplicity, we are interested in the decay of only one excitation in a system of two-level atoms, however, generalizations to multi-level atoms can be found in the literature[39]. Such system is represented by the state

$$|\Psi\rangle = \sum_{\mathbf{k}_\mu, \mathbf{k}_\nu} b_{\mathbf{k}}^{(g)}(t)|g_1 g_2 \cdots g_N\rangle|1_{\mathbf{k}}\rangle + \sum_{j=1}^N b_j^{(e)}(t)|g_1 g_2 \cdots e_j \cdots g_N\rangle|0\rangle \tag{11}$$

where $\mathbf{k}_{\mu(\nu)}$ is the sum over the guided (radiated) modes, $b_{\mathbf{k}}^{(g)}$ is the probability amplitude of all the atoms being in the ground state and one excitation in the $\mathbf{k}$-th mode of the field, and $b_j^{(e)}$ is the probability amplitude of having zero excitation in the field and an excitation in the $i$-th atom. Assuming that we start the cycle with the excitation in the atoms, i.e., $b_{\mathbf{k}}^{(g)}(0) = 0$, we can write the Schrödinger equation in the Markov approximation for the coefficients $b_i^{(e)}(t)$ in a matrix form as ref. [37]

$$\dot{\mathbf{B}}(t) = -\Gamma \mathbf{B}(t) \tag{12}$$

where $\mathbf{B}(t)$ is a vector with entries given by the $b_i^{(e)}(t)$, and $\Gamma$ is a non-hermitian symmetric matrix with entries $2\Gamma_{ij} = \gamma_{ij} + \mathrm{i}\Omega_{ij}$, representing the couplings between the $i$-th and $j$-th atoms calculated from the optical nanofiber modes, radiated and guided. The eigenvalues $\eta_\alpha$ of Eq. (12) give the possible decay rates of the system. These are the collective states mentioned in Eq. (1). The eigenvectors form a basis $\{|B_\alpha\rangle\}$ that allows us to write the state of the system as

$$|\Psi\rangle = \sum_{\mathbf{k}_\mu, \mathbf{k}_\nu} b_{\mathbf{k}}^{(g)}(t)|g_1 g_2 \cdots g_N\rangle|1_{\mathbf{k}}\rangle + \sum_{\alpha=1}^N c_\alpha e^{-\eta_\alpha t}|B_\alpha\rangle|0\rangle \tag{13}$$

where the coefficients $c_\alpha$ are given by the initial state. In contrast with Eq. (1), here we have also included the states with one excitation in the field.

Following this approach, the many-body problem, of calculating the decay of one excitation distributed among $N$ interacting atoms, becomes an eigenvalue problem in a Hilbert space of dimension $N^2$ instead of $2^N$. This speeds the calculations, allowing us to compute the decay rate of the system with Monte Carlo simulations for a large $N$ in random positions.

The electromagnetic field operator for the guided modes is ref. [25]

$$\hat{\mathbf{E}}_{\mathrm{guided}}^{(+)} = i \sum_{fp} \int_0^\infty \mathrm{d}\omega \sqrt{\frac{\hbar\omega\beta'}{4\pi\varepsilon_0}}\, \hat{a}_\mu \mathbf{e}^{(\mu)} e^{-i(\omega t - f\beta z - p\phi)}. \tag{14}$$

The formal solution of the Heisenberg equation for $\hat{a}_\mu(t)$ in the Markov and rotating-wave approximation is

$$\hat{a}_\mu(t) = \hat{a}_\mu(t_0) + 2\pi \sum_j G_{\mu j}^* \delta(\omega - \omega_0)\hat{\sigma}_j(t), \tag{15}$$

The substitution of this expression into Eq. (14) gives the guided field operator as a function of the dipole operators.

Assuming that the guided modes are initially empty and that all the dipoles are oriented in the $z$ direction and at the same distance from the ONF, the intensity of the guided field as a function of the atomic dipole operators is

$$\left\langle \hat{\mathbf{E}}_{\mathrm{guided}}^{(-)} \hat{\mathbf{E}}_{\mathrm{guided}}^{(+)} \right\rangle = |\mathcal{E}(r)|^2 |\mathrm{d}(t)|^2, \tag{16}$$

where

$$\mathrm{d}(t) = \sum_j e^{i(\beta z_j + \phi_j)} b_j^{(e)}, \tag{17}$$

$$|\mathcal{E}(r)|^2 = \frac{2\hbar\omega_0}{n_{\mathrm{eff}} c\varepsilon_0} \frac{\gamma_{1\mathrm{D}}(r)}{A_{\mathrm{eff}}(r)}, \tag{18}$$

considering $\gamma_{1\mathrm{D}}(r) = \gamma_{ii}^{(1\mathrm{D})}(r)$ from Eq. (8) and $A_{\mathrm{eff}(z)}(r) = \left| n_{\mathrm{eff}} e_z^{(\mu_0)}(r) \right|^{-2}$ to be the effective mode area of the $z$ component of the electric field[25]. Equation (18) relates the total radiated power into the waveguide with the energy radiated per unit time, i.e., $I(r)A_{\mathrm{eff}(z)}(r) = \hbar\omega_0\gamma_{1\mathrm{D}}(r)$, where $I(r)$ is the intensity of the radiated field.

Equation (16) shows that the measured intensity corresponds to the one produced by $N$ classical dipoles with different phases, different positions, and amplitudes given by the probability of being in the excited state $b_j^{(e)}$[40].

**Theoretical methods.** We use Monte Carlo simulations, randomly positioning $N$ atoms around the ONF. The position of each atom is given in cylindrical coordinates by $\mathbf{r}_i = (r_0, \phi_i, z_i)$, where $r_0 = (240 + 30)$ nm, $\phi_i \in [0, 2\pi]$, and $z_i$ is obtained from a Gaussian distribution with a FWHM of 200 μm, determined by the atomic cloud size. The radial position of the atoms is fixed, determined by the experimental procedure of repumping the atoms close to the nanofiber surface. In our case, all the atoms are at a constant radial position of 30 nm away from the surface of an ONF of 240 nm radius, with $\gamma_{1\mathrm{D}}/\gamma_0 \approx 0.13$. This is a good approximation given the narrow radial distribution of the atoms (~5 nm), as explained in the experimental methods.

The initial state will depend on the amplitude and phase of the excitation beam. We assume that the initial state corresponds to a superposition of all the atoms in the ground state except one with an induced atomic dipole. The initial phase between the atoms depends on their position; assuming an excitation pulse with a wave vector perpendicular to the fiber, each atom initial phase can be calculated from its coordinates. For each random realization, we solve Eq. (12) and calculate the intensity of the guided field, Eq. (16). We use these results to take the mean of the intensity of the guided field as a function of time. Typically, 100,000 realizations are required to converge to a level of precision higher than what it is visible in Figs. 2 and 3. If the mean of the intensity guided field is normalized, there is no dependence on the amplitude of the initial induced dipole in the weak excitation limit.

There is a correspondence between super-radiance (sub-radiance) configurations and constructive (destructive) interference of the field emitted by the dipoles into the ONF (see Eq. (17)); meaning that super-radiant configurations contributes more than sub-radiant configurations when taking the mean over all the realizations for an electric field detected through the ONF (Eq. (16)).

The theoretical model prediction for different dipole moment orientations relative to the ONF[25] qualitatively agrees with the observed experimental behavior: The long-term sub-radiance disappears on our signal-to-background-ratio window when exciting with vertically polarized light (see of Fig. 2c). A sensitivity analysis to the ONF radius shows no significant changes in the predictions up to a ±10 nm variation.

**Data availability**. The data that support the findings of this study are available from the authors on reasonable request.

Published online: xx xxx 2017

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

## Acknowledgements

We are grateful to A. Asenjo-Garcia, H. J. Carmichael, D.E. Chang, J. P. Clemens, M. Foss-Feig, M. Hafezi, B.D. Patterson, W.D. Phillips, and P.R. Rice for the useful discussions. We give special thanks to P. Zoller who besides discussing the topic of the paper helped us improve the manuscript. This research is supported by the National Science Foundation of the United States (NSF) (PHY-1307416); NSF Physics Frontier Center at the Joint Quantum Institute (PHY-1430094); the USDOC, NIST, Joint Quantum Institute (70NANB16H168); and the Office of the Secretary of Defense of the United States, Quantum Science and Engineering Program.

## Author contributions

P.S., F.K.F., L.A.O. and S.L.R. conceived the project. P.S. realized the measurements. P.B.-B. and P.S. developed the theoretical model. All authors discussed the results, contributed to the data analysis, and worked together on the manuscript.

## Additional information

**Competing interests:** The authors declare no competing financial interests.

