## [Peer Review File · Nature Communications]

Reviewers' comments:

Reviewer #3 (Remarks to the Author):

The authors have satisfactorily replied to most of my comments. I believe the manuscript should be accepted in Nature Communications once the authors address the following concerns:

1. The sentence "For free-space coupling, collective states emerge when all the atoms lie within a wavelength" is not accurate, as the authors can readily check by diagonalizing their matrix Γ (appearing in Eq. 12), for a chain of atoms separated by a distance of half a wavelength, or even a wavelength of each other (and thus totalling a distance way larger than a wavelength). In my opinion, it would be more accurate to say that for atom-atom separations beyond a (very) few wavelengths, collective effects are not significant, in contrast to what happens for interactions mediated by the waveguide.

2. To my comment: "Line 113: The expression for the optical depth reads $n_{\text{eff}}OD = N\gamma^{1D}/\gamma_0$. However, shouldn't it be $n_{\text{eff}}OD = N\gamma^{1D}/\gamma^{\text{rad}}?$ ", the authors have replied "The original expression $n_{\text{eff}}OD = N\gamma^{1D}/\gamma_0$ is correct. See Ref. [8] Chapter 1 for details.". However Ref. [8] is a PRL article where this expression does not appear.

3. I do not understand the response of the authors to one of my questions, and I would like them to clarify this point further, if possible. In particular, the authors consider that atoms are only polarized along z , and I wondered in my previous report why this is the case, as the guided mode field can have any polarization and thus would drive other atomic transitions. The response of the authors to this question is the following: "[...] the system is not free from polarization mixing, given the complicated polarization of a guided mode. However, this is taken into account in calculation of atom-atom coupling mediated by the guided mode of between two atomic dipole oriented along the z -axis." How can the complex polarization of the guided mode be taken into account just considering atoms polarized along the z -axis?

We thank the reviewer for the careful reading of the manuscript and for asking questions that help clarify significantly our manuscript.

Reviewers' comments:

Reviewer #3 (Remarks to the Author):

1. The sentence “For free-space coupling, collective states emerge when all the atoms lie within a wavelength” is not accurate, as the authors can readily check by diagonalizing their matrix Γ (appearing in Eq. 12), for a chain of atoms separated by a distance of half a wavelength, or even a wavelength of each other (and thus totalling a distance way larger than a wavelength). In my opinion, it would be more accurate to say that for atom-atom separations beyond a (very) few wavelengths, collective effects are not significant, in contrast to what happens for interactions mediated by the waveguide.

We agree with the referee and we appreciate the comment. The sentence, which had the purpose of emphasizing the difference between free-space and one-dimensional dipole-dipole coupling, is indeed inaccurate. It has been changed, and now reads:

“For free-space coupling, collective states emerge for atom-atom separations smaller than a few wavelengths.”

2. To my comment: “Line 113: The expression for the optical depth reads $n_{eff}OD=N\gamma_{1D}/\gamma_0$. However, shouldn't it be $n_{eff}OD=N\gamma_{1D}/\gamma_{rad}$?”, the authors have replied “The original expression $n_{eff}OD=N\gamma_{1D}/\gamma_0$ is correct. See Ref. [8] Chapter 1 for details.” However Ref. [8] is a PRL article where this expression does not appear.

We are sorry for this error. It should have said Ref. [11]. This is a review article called “Optical Nanofibers: A New Platform for Quantum Optics.” *Advances in Atomic, Molecular, and Optical Physics*, **66** 439 (2017).

A simple derivation can be obtained by writing the atomic decay rate in vacuum and into the guided mode as:

$$\gamma_0 = \frac{2\pi}{\sigma_0} \frac{d^2}{\pi\epsilon_0\hbar c} \omega$$

$$\gamma_{1D} = n_{eff} \frac{2\pi}{A_{mod}} \frac{d^2}{\pi\epsilon_0\hbar c} \omega$$

where σ_0 and A_{mod} are the resonant atomic cross section and the effective area of the guided mode. The decay rate of an atom in a dielectric medium is proportional to the index of refraction of the medium (or the inverse of the group velocity). A recommended reference on the subject is *Phys. Rev. A* **80**, 011810R (2009). The ratio of these two quantities multiplied by the number of atoms in the mode (N) leads to

$$N \frac{\gamma_{1D}}{\gamma_0} = n_{eff} N \frac{\sigma_0}{A_{mod}} = n_{eff} OD$$

where the optical density is given by the number of atoms times the ratio of the atomic cross section and the mode area ($OD = N \frac{\sigma_0}{A_{mod}}$).

3. I do not understand the response of the authors to one of my questions, and I would like them to clarify this point further, if possible. In particular, the authors consider that atoms are only polarized along z , and I wondered in my previous report why this is the case, as the guided mode field can have any polarization and thus would drive other atomic transitions. The response of the authors to this question is the following: “[...] the system is not free from polarization mixing, given the complicated polarization of a guided mode. However, this is taken into account in calculation of atom-atom coupling mediated by the guided mode of between two atomic dipole oriented along the z -axis.” How can the complex polarization of the guided mode be taken into account just considering atoms polarized along the z -axis?

We understand the concern of the referee, and we regret that we did not address it properly in the original submission since this is a subtle and important point.

The external probe is linearly polarized along z (when we excite with H), thus all the weakly driven atoms acquire an induced dipole moment in the z direction. This is the initial condition for all the atomic dipole moments in the system. The guided electromagnetic field that mediates the atom-atom interactions has indeed a complicated polarization. However, the calculations are simplified because the atom-light interaction is given by the dot product between the electric field and the atomic dipole, and the latter is set from the initial conditions (the probe polarization). The detailed calculation of the atom-atom coupling for atoms aligned along the z -axis (and all other possible combinations of alignment) is well described in Ref. [25] (of the newer version of the manuscript, Phys. Rev. A, **72** 063815, (2005)), and we reproduce the same calculations.

The scenario would be different if, for example, all the atoms were initially in the excited state, a more familiar case in quantum optics known as superfluorescence. In that case, after the system starts decaying atoms will acquire an induced dipole moment in the direction of polarization of the guided mode electric field. The same is true for any initial state configuration with zero induced atomic dipole moment. This would definitely complicate the calculations, however the infinite range atom-atom interaction would still be present. The inset in Fig. 2 shows how the collective decay signal looks for two different initial atomic states: the initial atomic dipoles oriented along z , and in a mixture of r (radial) and ϕ (azimuthal).

To clarify this point we have modified the manuscript. It now reads:

“For simplicity, here we focus only on the case where atoms are prepared in an initial state with induced atomic dipoles oriented along the ONF (z -axis). This is true for atoms weakly driven by an external probe polarized along z .”

REVIEWERS' COMMENTS:

Reviewer #3 (Remarks to the Author):

The authors have satisfactorily replied to my two first comments. Regarding the third point, I still do not agree. I think assuming that atoms are polarized along z is an approximation, as the atomic dipole responds to the total field impinging on the atom, which is both the external probe and also the scattered field. I am willing to acknowledge that the scattered field through the fiber is small compared to the external field and thus the atoms can be considered to be two-level emitters (as done in Ref[25], where the complex polarization structure of the atoms is not taken into account from the beginning). At the same time, performing the calculations including all polarizations would be very complicated.

Referee's comment:

The authors have satisfactorily replied to my two first comments. Regarding the third point, I still do not agree. I think assuming that atoms are polarized along z is an approximation, as the atomic dipole responds to the total field impinging on the atom, which is both the external probe and also the scattered field. I am willing to acknowledge that the scattered field through the fiber is small compared to the external field and thus the atoms can be considered to be two-level emitters (as done in Ref[25], where the complex polarization structure of the atoms is not taken into account from the beginning). At the same time, performing the calculations including all polarizations would be very complicated.

We thank the referee for the comment and we take it as an opportunity to further explain the consequences and validity of the approximation made. The paragraph now reads:

“For simplicity, here we focus only on the case where atoms are regarded as two level systems prepared in an initial state with induced atomic dipoles aligned along the ONF (z -axis). This is a reasonable approximation for atoms weakly driven by an external probe polarized along z . In a realistic scenario, the light scattered by the fiber and by the multi-level internal structure of the atoms can mix the light polarization. The computation of such a system becomes cumbersome and only contributes to correction to the dominant effect. A description given by two level atoms aligned along the z -axis allows us to quantitatively capture the physical phenomena while keeping the mathematical description simple.”